# GRAPH OF AGENTS: PRINCIPLED LONG CONTEXT MODELING BY EMERGENT MULTI-AGENT COLLABORATION

## ABSTRACT

As a model-agnostic approach to long context modeling, multi-agent systems can process inputs longer than a large language model's context window without retraining or architectural modifications. However, their performance often heavily relies on hand-crafted multi-agent collaboration strategies and prompt engineering, which limit generalizability. In this work, we introduce a principled framework that formalizes the model-agnostic long context modeling problem as a compression problem, yielding an information-theoretic compression objective. Building on this framework, we propose Graph of Agents (GoA), which dynamically constructs an input-dependent collaboration structure that maximizes this objective. For Llama 3.1 8B and Qwen3 8B across six document question answering benchmarks, GoA improves the average $F_1$ score of retrieval-augmented generation by 5.7% and a strong multi-agent baseline using a fixed collaboration structure by 16.35%, respectively. Even with only a 2K context window, GoA surpasses the 128K context window Llama 3.1 8B on LongBench, showing a dramatic increase in effective context length.

## 1 INTRODUCTION

From analyzing entire codebases to synthesizing evidence across multiple documents, critical AI applications require reasoning over long contexts. Yet, modern large language models (LLMs) remain fundamentally constrained by their context windows, the maximum number of tokens they can reliably process. When inputs exceed those seen during training, performance quickly deteriorates (Anil et al., 2022; Zhou et al., 2024). Further, some architectures cannot accept inputs beyond their pre-trained length due to constraints such as fixed positional encodings (Press et al., 2022). While retraining on longer inputs seems like a direct solution, it is often infeasible due to prohibitive computational costs and data scarcity. This mismatch between model capabilities and practical needs hinders LLM deployment in complex real-world tasks.

Post-hoc modification methods offer appealingly simple and training-free alternatives. For example, position embedding interpolation (Chen et al., 2023b) extends pre-trained position encodings to longer sequences. Vasylenko et al. (2025) propose a sparse attention method that preserves distributional characteristics of the attention weights by restricting information flow to fixed patterns in longer sequences. However, they often do not enhance the long context modeling ability, leaving a gap between the claimed context window (maximum input length) and the effective context window (range where the model reliably performs) (Liu et al., 2024; Hsieh et al., 2024). Addressing long context challenges requires paradigms beyond simply increasing input length.

Multi-agent systems offer a promising alternative for long context modeling by leveraging auxiliary LLMs to restructure and compress the input. A prominent paradigm is to orchestrate agents in structured workflows. For example, in Chain-of-Agents (CoA) (Zhang et al., 2024), agents collaborate in a sequential chain to progressively summarize the input (cf. Figure 1 (Bottom)). LongAgent (Zhao et al., 2024a) employs a leader agent to dispatch tasks to a team of specialized member agents, manually defining a collaborative workflow

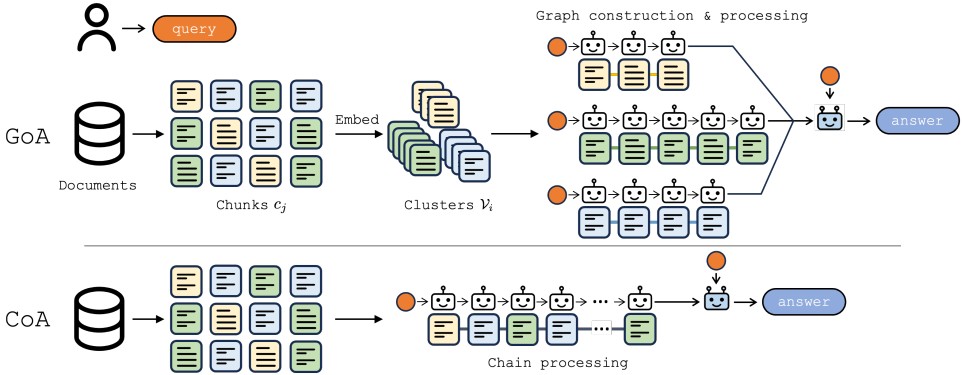

Figure 1: A comparison of multi-agent systems for long context modeling, where worker agents (🤖) process distinct text chunks and collaborate to generate a compressed summary for a manager agent (🤖). **Top: Graph of Agents:** Our proposed method generates an input-dependent collaboration structure. First, agents are grouped into clusters based on the semantic similarity of their assigned chunks. Then, within each cluster, a collaboration structure is adaptively determined to generate an optimal partial summary tailored for each input and query. This allows for flexible and parallelizable compression. **Bottom: Chain-of-Agents.** A baseline approach where worker agents collaborate in a prescribed serial order for every input. This static structure can be suboptimal when relevant information is not organized linearly in the source document.

based on pre-assigned roles. Such strategies have been shown to effectively handle inputs far beyond a model's context window, outperforming post-hoc modification methods (Zhang et al., 2024; Lee et al., 2024). However, current multi-agent approaches rely heavily on hand-crafted designs, such as predefined roles, fixed collaboration structures, and task-specific prompts (e.g., a serial communication chain in CoA), that may not generalize across diverse tasks.

In this work, we design a principled multi-agent system where an input-dependent optimal collaboration structure emerges from the statistical structure of an input. To this end, we formalize the long context modeling problem as a compression problem, yielding a natural information-theoretic objective. Building on this foundation, we propose Graph of Agents (GoA), which models multi-agent collaboration as a dynamic graph tailored to each input (cf. Figure 1 (Top)). In GoA, agents (nodes) responsible for distinct text chunks are first grouped into clusters based on semantic similarity. Then, within each cluster, a communication structure (edges) is determined greedily by selecting the next chunk most relevant to the query given the evolving summary. Crucially, this adaptive graph structure guided by the information-theoretic objective allows GoA to generalize across diverse tasks, from locating knowledge from single-document to complex multi-document reasoning, without requiring task-specific hand-crafted workflows. As a result, GoA improves the average $F_1$ score of retrieval-augmented generation by 5.7% and CoA by 16.35%, respectively, for Llama 3.1 8B and Qwen3 8B across six single- and multi-document question-answering datasets.

Our contribution can be summarized as follows: **1)** We formalize long context modeling from the perspective of compression, providing unique insights into existing long context modeling approaches and guiding a principled multi-agent system design; **2)** We develop a training-free multi-agent system, GoA, where the input-dependent collaboration structure emerges to maximize the principled long context compression objective; **3)** GoA effectively handles inputs longer than the context window, notably outperforming a 128K context window Llama 3.1 8B with only a 2K context window model on LongBench.

## 2 BACKGROUND: MODEL-AGNOSTIC APPROACH FOR LONG CONTEXT MODELING

We address the problem of applying an LLM $\hat{P}$ with context window $T_{\max}$ to inputs exceeding this limit. When a prompt $\mathrm{prompt}(x, q)$ constructed from an input $x$ and query $q$ is too long (i.e., $|\mathrm{prompt}(x, q)| >$

$T_{\max}$), the model cannot directly compute $\hat{P}(y \mid \texttt{prompt}(x,q))$. In this setting, the model-agnostic long context modeling aims to find a compressed representation $(\tilde{x}, q) \triangleq (g(x;q), q)$ with a compression function $g$ summarizing input $x$ conditioned on a query $q$, ensuring that $|\texttt{prompt}(g(x;q),q))| \leq T_{\max}$. For brevity, we omit the explicit dependence on $q$ in $g$ throughout the paper (i.e., $g(x) \triangleq g(x;q)$).

In the following, we review two popular directions for model-agnostic long context modeling that design $g$ that effectively preserves the information necessary to answer the query. Both approaches process long inputs by dividing them into smaller chunks. Accordingly, without loss of generality, we assume that $x$ can be divided into $l$ number of chunks $(c_1, c_2, \cdots, c_l)$ with $|c_i| = L$ for all $i \in [l] \triangleq \{1, \cdots, l\}$.

**Retrieval-augmented generation (RAG)** implements compression by retrieving relevant chunks within the context window $T_{\max}$ (Lewis et al., 2020). Specifically, RAG first ranks the chunks based on their proximity to the query $q$ (e.g., lexical matching (Robertson et al., 1995)) and then selects the top $\kappa$ relevant chunks such that $\kappa = \max\{a \in \mathbb{N} \mid a \cdot |L| \leq T_{\max}\}$: $\tilde{x}^{\text{RAG}} = [c_{\pi^{\text{RAG}}(1)}, \cdots, c_{\pi^{\text{RAG}}(\kappa)}]$ where $\pi^{\text{RAG}}(i)$ is the index of the $i$-th closest chunk to the query $q$. While conceptually simple, RAG-based approaches have been shown to be highly effective for long context modeling, often outperforming methods that naively attend to the entire input, especially for knowledge-intensive tasks like document-based question answering (Sarthi et al., 2024; Zhao et al., 2024b). However, this filtering-style approach risks discarding information that is not directly similar to the query but nevertheless essential for comprehensive reasoning and understanding.

**Multi-agent systems** iteratively summarizes $x$ by using multiple collaborating LLMs. As a prominent example, we consider the Chain-of-Agents (CoA) framework (Zhang et al., 2024) for its broad applicability. Unlike many systems that rely on highly specialized agents with hand-crafted roles and prompts, CoA employs a chain of *homogeneous worker agents*, each performing a generic summarization task to collaboratively synthesize the document. Specifically, each worker agent $W_i$ produces a communication unit $z_i$, which is a summary of a chunk $c_i$ and the previous communication unit $z_{i-1}$ (with $z_0$ being an empty string), and passes $z_i$ to the next worker $W_{i+1}$. Given $(x, q)$, this process is formalized as

$$z_i = W_i(c_i, z_{i-1}, q), \quad \text{for } i = 1, 2, \cdots, l, \tag{1}$$

where $W_i$ involves a structural prompt and an inference by an LLM (cf. §E.3). The final worker output $z_l$ is passed to the LLM manager $M$ for producing an output $\hat{y}$ by $\hat{y} = M(z_l, q)$ as described in §E.3.

While the CoA paper shows that its sequential communication of providing the contextual information $z_{i-1}$ to the next agent is key to outperforming parallel systems (Zhao et al., 2024a; Lee et al., 2024), this design is fundamentally heuristic. Its effectiveness is not grounded in a principled objective, which obscures its potential limitations and prevents generalization. Similarly, many existing multi-agent systems rely on hand-crafted rules, such as assigning agents specialized roles for atomic tasks like question decomposition or fact verification (Zhao et al., 2025) or employing a supervising agent to manage other specialists within a fixed workflow (Zhao et al., 2024a). This lack of a theoretical foundation makes it hard to understand the multi-agent system's behavior and its performance unreliable across diverse tasks and inputs.

## 3 COMPRESSION PERSPECTIVE ON LONG CONTEXT MODELING

The heuristic nature of existing multi-agent systems highlights a fundamental gap: they lack a formal criterion for optimal compression. Our goal is to derive this criterion from first principles, which will provides the foundation of our novel method in §4. The natural goal of the compression function $g$ is to produce a representation $\tilde{x}$ that maximizes the performance of the LLM $\hat{P}$. This can be formulated as an optimization problem $\max_g \mathbb{E}_{(X,Q,Y) \sim P}[\log \hat{P}(Y|g(X), Q)]$ where $(X, Q, Y)$ are random variables of input, query and output with $P$ being the data generating distribution.

The main limitation of this objective is its lack of predictive power; it can only evaluate a compression function retrospectively, not guide its design. Therefore, we consider the case where the LLM is a Bayes optimal

predictor (i.e., it models the underlying posterior probability almost everywhere (Jeon et al., 2024)). This idealization isolates the information loss attributable solely to the compression method, while remaining agnostic to characteristics of any particular model. Under this condition, the following proposition establishes a natural information-theoretic objective for compression.

**Proposition 1.** *For the Bayes optimal predictor $\hat{P}$, the following equivalence relationship holds:*

$$\mathbb{E}_{(X,Q,Y)\sim P}\left[\log \hat{P}(Y|g(X),Q)\right] \iff I(Y;g(X),Q), \tag{2}$$

*where $I(X;Y) \triangleq \mathbb{E}\left[\log \frac{P(X,Y)}{P(X)P(Y)}\right]$ is the mutual information.*

The proof is based on an elementary result about the Bayesian posterior (see §A.1). Proposition 1 provides a powerful and intuitive objective that reframes the goal of compression for long context modeling as preserving information about the final answer. This result extends to imperfect LLMs, where the mutual information objective remains a valid compression metric, up to a tolerance term reflecting the LLM's error (cf. Proposition 2 in §B). Unlike prior work that relies heavily on empirical benchmarking, this perspective yields a predictive criterion for evaluating compression $g(X)$. While this provides a principled criterion, the objective's intractability (Poole et al., 2019) and dependence on the unknown output $Y$ prevent its direct use for guiding multi-agent collaboration. Building upon this foundation, we derive a tractable objective in the next section, from which we construct a principled multi-agent system designed for its optimization.

## 4 GRAPH OF AGENTS

We introduce Graph of Agents (GoA), a framework that models multi-agent collaboration as an input-dependent graph to optimize the mutual information objective in (2). In this graph, each agent is assigned a text chunk and represents a node. The edges, which act as communication channels, are established based on the statistical relationship between chunks. In the following, we develop GoA by addressing three key algorithmic designs: **1)** Determine a proper family of graph structures for long context modeling; **2)** Formulate the intractable mutual information objective in (2) as a tractable surrogate; **3)** Generate an input-dependent graph that maximizes this tractable objective. We present pseudocode of GoA in Algorithm 1.

### 4.1 INDUCTIVE BIAS ON THE GRAPH STRUCTURE

Introducing proper inductive biases is an effective way to reduce the complexity of AI system design with desired qualities. To enable a fast and flexible communication structure, we introduce a structural inductive bias by constraining the agent collaboration topology to that of a *linear forest*. A linear forest $\mathcal{F}$ is a graph whose connected components $\{\mathcal{G}_i\}_{i\in[k]}$ are all path graphs. Formally, this means that the set of all agents (representing chunks $\{c_1, c_2, \cdots, c_l\}$[1]) is partitioned into disjoint vertex sets $\mathcal{V}_1, \cdots, \mathcal{V}_k$. Each subgraph $\mathcal{G}_i = (\mathcal{V}_i, \mathcal{E}_i)$ is a path graph, where the directed edge set $\mathcal{E}_i$ is defined by an ordering $\pi_i$ of the agents in $\mathcal{V}_i$: $\mathcal{E}_i = \{(\pi_i(j), \pi_i(j+1))|j \in [|\mathcal{V}_i|]\}$, where $\pi_i(j)$ represents the $j$-th worker in $\mathcal{G}_i$. We denote the family of all possible linear forests satisfying these conditions as $\mathbb{F}$.

GoA's primary goal is to instantiate an optimal input-dependent linear forest $\mathcal{F}^*(x, q) \in \mathbb{F}$. Before describing how GoA finds this optimal structure in §4.3, we first explain the collaboration mechanism for any given forest $\mathcal{F} \in \mathbb{F}$. Within each path graph $\mathcal{G}_i \in \mathcal{F}$, worker agents $\{W_{\pi_i(j)}\}_{j=1}^{|\mathcal{V}_i|}$ collaborate sequentially to compress the chunks assigned to them (corresponding to $\mathcal{V}_i$). Each worker agent $W_{\pi_i(j)}$ takes its assigned chunk $c_{\pi_i(j)}$ and the summary from the previous agent $z_{\pi_i(j-1)}$ to produce an updated summary $z_{\pi_i(j)}$:

$$z_{\pi_i(j)} = W_{\pi_i(j)}\left(c_{\pi_i(j)}, z_{\pi_i(j-1)}, q\right), \quad \text{for } j = 1, 2, \cdots, |\mathcal{V}_i|, \tag{3}$$

---

[1]As in CoA (Zhang et al., 2024), we construct a chunk by sequentially adding paragraphs until it reaches the context window. We deliberately chose this generic chunking strategy to focus on the effects of the communication topology, leaving the exploration of more general chunking methods as future work.

where $W_{\pi_i(j)}$ performs an LLM inference prompted to summarize its inputs $(c_{\pi_i(j)}, z_{\pi_i(j-1)})$ for answering the query $q$ (cf. §E.3 for the prompt). The final output $z_{\pi_i(|\mathcal{V}_i|)}$ summarizes chunks in $\mathcal{V}_i$.

After all final summaries $\{z_{\pi_i(|\mathcal{V}_i|)}\}_{i\in[k]}$ are generated, a manager agent $M$ synthesizes them to produce the final answer. The total length of these summaries is designed to fit within $T_{\max}$ by properly setting a maximum output token limit for each worker. The manager's operation is defined as:

$$y = M\left(\oplus_{i=1}^k z_{\pi_i(|\mathcal{V}_i|)}, q\right), \tag{4}$$

where $\oplus_{i=1}^m a_i = a_1 \oplus a_2 \oplus \cdots \oplus a_m$ is the concatenation of $(a_1, a_2, \cdots, a_m)$ and $M$ performs an LLM inference prompted to synthesize an answer from the provided summaries (cf. §E.3 for the prompt).

The linear forest structure in GoA provides two key advantages over single-chain methods (e.g., CoA in (1)): _Parallelization_: Inferences across the different subgraphs $(\mathcal{G}_1, \cdots, \mathcal{G}_k)$ can be run in parallel, significantly reducing latency. _Flexibility_: The input-dependent paths $\pi_i$ are determined dynamically for each input, creating more effective data-driven collaboration than a static collaboration structure (e.g., a serial chain).

### 4.2 TRACTABLE COMPRESSION OBJECTIVE FORMULATION

Our goal is to design a compressor $g$ that maximizes the mutual information objective $I(Y; g(X), Q)$ (cf. (2)). Direct optimization is infeasible due to its dependence on $Y$. Recent work (Liu et al., 2025), however, reveals a strong connection between $I(Y; g(X), Q)$ and a tractable surrogate $I(Q; g(X))$: for particular answer $y$ of a query $q$ and an input $x$, the log-odds of the answer $\log \frac{P(y|q,x)}{1-P(y|q,x)}$ are strongly correlated with the pointwise mutual information $PMI(q, x) = \log \frac{P(q,x)}{P(q)P(x)}$. Intuitively, if a question and input co-occur frequently, the input is more likely to contain the information needed to answer the question. This motivates replacing $I(Y; g(X), Q)$ with $I(Q; g(X))$ (see derivation in §A.2).

Although $I(Q; g(X))$ is tractable in principle, estimating the mutual information remains computationally challenging (Poole et al., 2019). To address this, we exploit the relationship between embedding distance and the distributional similarity (i.e., pointwise mutual information (PMI)). For word embeddings, inner products approximate PMI: $e(w)^T e(w') \approx PMI(w, w')$ (Levy & Goldberg, 2014; Arora et al., 2016). Intuitively, since word embedding spaces capture co-occurrence statistics, proximity in embedding space reflects distributional similarity. The same intuition extends to sentence embeddings trained with contrastive losses that maximize similarity between related documents while minimizing similarity to irrelevant ones. In particular, InfoNCE, which is a widely used contrastive objective, has been shown to approximate PMI at optimality (Oord et al., 2018; Zhang et al., 2023). Specifically, for chunks $c$ and $c'$, it holds that

$$\text{emb}(c)^T \text{emb}(c') = \log \frac{P(c|c')}{P(c)} + \gamma(c') = PMI(c; c') + \gamma(c'), \tag{5}$$

where $\gamma$ is an arbitrary function and $\text{emb}$ denotes an embedding model trained with InfoNCE.

We therefore approximate the intractable optimization problem $\max_g I(Y; g(X), Q)$ by finding the compressed representation $g(x; q)$ maximizing the semantic similarity $\max_{g(x;q)} \text{emb}(g(x; q))^T \text{emb}(q)$ for all $(x, q)$ pairs. This tractable surrogate allows us to operationalize input-dependent compression while preserving theoretical grounding in the principled mutual information objective.

### 4.3 DYNAMIC INPUT-DEPENDENT GRAPH CONSTRUCTION

Combining §4.1 and §4.2, GoA instantiates a specific linear forest that generates a compressed representation most semantically aligned with the query. Specifically, for each $(x, q)$, we aim to find the linear forest

$$\mathcal{F}^*(x, q) \in \underset{\mathcal{F}:\text{Linear Forest}}{\arg\max} \ \text{sim}\left(\text{emb}(q), \text{emb}(\tilde{x}_{\mathcal{F}}^{\text{GoA}})\right), \tag{6}$$

where $\tilde{x}_{\mathcal{F}}^{\text{GoA}}$ is the final concatenated output from workers of GoA under $\mathcal{F}$ and sim is the cosine similarity.

Directly solving the combinatorial optimization problem in (6) is impractical since evaluating objective value of $\mathcal{F}$ requires executing all worker communication chains via sequential LLM inferences (cf. (3)). Instead, we design a greedy graph construction procedure that clusters chunks and sequentially builds paths within each cluster to maximize query similarity.

*Node construction.* The linear tree prevents a worker from leveraging contextual information from chunks in different subgraphs. To minimize its impact, we group semantically related chunks together with a clustering algorithm that finds $k$ clusters $\{\mathcal{V}_1, \cdots, \mathcal{V}_k\}$ in the embedding space. This reduces redundancy across subgraphs while preserving local coherence.

*Edge construction.* Within each cluster $\mathcal{V}_i$, we

---

**Algorithm 1.** Graph of Agents.

**Input** Input $x$, query $q$, cluster size $k$
**Output** Prediction $\hat{y}$
**Require** Embedding model, clustering algorithm

1: Decompose $x$ into chunks $(c_1, c_2, \cdots, c_l)$
2: Construct $k$ number of clusters $\mathcal{V}_1, \cdots, \mathcal{V}_k$ in the embedding space
3: **for** $i = 1, \cdots, k$ **do**           ▷ *Parallelizable*
4:     Initialize $z_{\pi_i(0)} = \emptyset$
5:     **for** $j = 1, \cdots, |\mathcal{V}_i|$ **do**
6:         Determine a next chunk $c_{\pi_i(j)}$ semantically closest to $q$ given a context $z_{\pi_i(j-1)}$ with (7)
7:         Worker LLM generates $z_{\pi_i(j)}$ summarizing a chunk $c_{\pi_i(j)}$ and a context $z_{\pi_i(j-1)}$ with (3)
8:     **end for**
9: **end for**
10: Manager LLM generates a prediction $\hat{y}$ from outputs from all subgraphs $\{z_{\pi_i(|\mathcal{V}_i|)}\}_{i=1}^{k}$ with (4)

---

construct a path $\pi_i(1:|\mathcal{V}_i|)$ under which the final worker output is semantically closest to the query, aiming to maximize $\text{sim}\big(\text{emb}(q), \text{emb}(z_{\pi_i(|\mathcal{V}_i|)})\big)$ with respect to $\pi_i(1:|\mathcal{V}_i|)$. To this end, we implement a greedy algorithm that sequentially selects a chunk that maximally increases the similarity given the current context:

$$\pi_i(j) \in \underset{c \in \mathcal{V}_i, c \notin \pi_i(1:j-1)}{\arg\max} \text{sim}(\text{emb}(q), \text{emb}(z_{\pi_i(j-1)} \oplus c)), \quad j = 1, 2, \cdots, |\mathcal{V}_i|. \tag{7}$$

This greedy step progressively builds a summary that is increasingly aligned with the query's informational needs, ensuring that each chunk is selected based on its marginal utility given the context generated so far.

**Discussion.** In Appendix D, we discuss advantages of GoA's worker collaboration in (3) as a flexible contextual compressor. Compared to GoA, multi-agent systems without contextual compressors (e.g., Zhao et al. (2024a)) are fundamentally inferior at maximizing the compression objective since the lack of context prevents them from performing critical functions like disambiguating terms or eliminating redundancy. Also, RAG's filtering-style compression makes it inherently hard to understand the entire context unlike GoA.

## 5 EXPERIMENTS

### 5.1 SETUP

**Datasets.** We evaluate GoA using $F_1$ score on single and multi documents QA tasks on six question answering datasets from LongBench (Bai et al., 2024). Single-document QA datasets (NarrativeQA, Qasper, MultifiledQA) test reading comprehension of long stories (movie/book), academic papers, and Wikipedia with both yes/no/unanswerable and open-ended types of questions. Multi-document QA datasets (HotpotQA, 2WikiM, Musique) have open-ended questions requiring multi-hop reasoning (upto 5-hop questions) and therefore require understanding the entire context. The datasets range from 3.6K to 18K tokens, on average (see §E.4 for dataset details). We use generic prompts such as instructing only output formats to minimize any biases coming from hand-crafted prompts (see §E.3 for exact prompts used for each benchmark).

**Backbone LLMs & Baselines.** We use Llama 3.1 8B and Qwen3 8B as backbone LLMs for all methods. As baselines, we consider a RAG with a strong retriever (Xiao et al., 2024) with 300-word chunks, Chain-of-Agents (Zhang et al., 2024), and a vanilla base model. To evaluate performance on inputs much longer than

Table 1: Benchmark results on question answering tasks in LongBench. Scores are $F_1$ averages over three random seeds. Boldface highlights the best method for each context window.

| Base Model | Method | Single Doc QA | | | Multi Doc QA | | | |
| | | NarrativeQA | Qasper | MultifieldQA | Hotpot | 2WikiM | Musique | Average |
| --- | --- | --- | --- | --- | --- | --- | --- | --- |
| Llama 3.1 8B | Vanilla (2K) | 27.05 | 58.03 | 47.83 | 50.68 | **49.22** | 30.33 | 43.86 |
| | Vanilla (8K) | 34.82 | 48.25 | 58.44 | **63.82** | 46.06 | **37.35** | 48.12 |
| | Vanilla (128K) | 30.05 | 51.09 | 59.83 | 63.95 | 47.64 | 34.94 | 47.92 |
| | RAG (2K) | 32.68 | 51.65 | **54.71** | 46.18 | 47.46 | 26.37 | 43.18 |
| | RAG (8K) | **38.37** | 41.63 | **58.35** | 51.17 | 49.00 | 36.48 | 45.83 |
| | CoA (2K) | 25.90 | 59.32 | 49.76 | 44.27 | 32.02 | 19.96 | 38.54 |
| | CoA (8K) | 28.53 | 66.81 | 53.38 | 46.64 | 56.43 | 24.67 | 46.08 |
| | GoA (2K) | **34.11** | **66.20** | 51.17 | **52.88** | 49.00 | **38.66** | **48.67** |
| | GoA (8K) | 36.76 | **68.05** | 53.02 | 46.59 | **59.23** | 31.40 | **49.18** |
| Qwen3 8B | Vanilla (2K) | 9.74 | 60.35 | 41.11 | 52.68 | 40.34 | 21.66 | 37.65 |
| | Vanilla (8K) | 19.09 | 69.67 | 48.37 | 55.28 | 47.44 | 30.47 | 45.05 |
| | Vanilla+YaRN (128K) | 23.55 | 70.32 | 50.95 | 59.14 | 50.75 | 38.13 | 48.81 |
| | RAG (2K) | 9.65 | **76.06** | 45.89 | 54.37 | 47.04 | 21.43 | **42.41** |
| | RAG (8K) | 19.07 | **72.04** | 51.35 | 53.43 | 52.37 | 35.47 | 47.29 |
| | CoA (2K) | **17.52** | 56.96 | 43.23 | 44.94 | **49.69** | 24.49 | 39.47 |
| | CoA (8K) | **22.35** | 57.12 | 49.20 | 52.45 | 63.33 | 24.43 | 44.81 |
| | GoA (2K) | 14.51 | 57.59 | 42.12 | **56.79** | 46.28 | **35.06** | 42.06 |
| | GoA (8K) | 15.82 | 54.56 | 41.01 | 57.87 | **63.37** | **36.75** | **48.98** |

the available context, we test all methods under different context windows (2K, 8K, and 128K). Following the methodology in Bai et al. (2024), any input longer than the context window is truncated by removing content from its middle. Additional implementation details are deferred to §E.1.

**GoA Setup.** We use BGE-M3 (Chen et al., 2024a) that uses BERT (Devlin et al., 2019)-based architecture trained with the InfoNCE loss, matching our setting discussed in §4.2. We set the number of subgraphs $k = 4$ for all benchmarks, and find clusters with $k$-medoid clustering algorithm for its robustness to outliers.

## 5.2 BENCHMARK RESULTS

**Question Answering Tasks in LongBench.** Table 1 shows the results on the LongBench QA tasks. Our findings demonstrate the *superiority of GoA's flexible collaboration for handling long context data*. For both Llama 3.1 8B and Qwen3 8B, GoA consistently outperforms all baselines with the same context window. For instance, on Llama 3.1 8B with a 2K context, GoA achieves an average $F_1$ score of 48.67, a significant improvement over Vanilla (43.86), RAG (43.18), and CoA (38.54). Further, *GoA significantly extends the effective context window*, generally allowing a model with a shorter context window to outperform the vanilla model with a larger one. Notably, GoA on Llama 3.1 8B with an 2K window (48.67 $F_1$) outperforms the 128K context window model (47.92 $F_1$). This confirms that GoA serves as a powerful and efficient method for processing long context inputs without expensive retraining.

*Flexible collaboration is crucial for complex multi-hop reasoning.* Unlike GoA, CoA struggles on multi-document QA tasks like HotpotQA and Musique. This is because GoA's flexible collaboration guides an optimal information processing order, while the linear chain of CoA can be ill-suited for these non-sequential tasks. This leads to a significant performance gap between CoA and GoA on the multi-document QA tasks with a notable statistical significance $p \approx 0.006$ under the paired t-test. This is a stark contrast to CoA's strong performance on single-document QA tasks, where its serial communication chain aligns well with a natural linear information flow. Further, while RAG is shown to be very effective at single doc QA that requires to recall some factual information that directly answer the query, it also does not consistently improve the performance of the vanilla method (only five out of twelve cases, as opposed to nine cases in GoA).

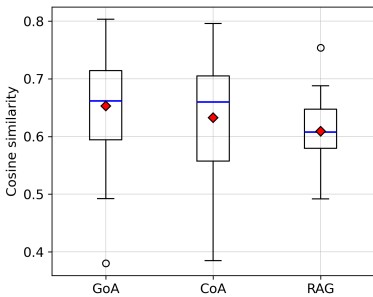

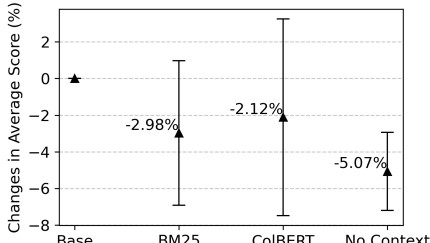

Figure 2: Proximity of a summarized input to a query measured by the cosine similarity in the embedding space. The box plot describes the interquartile range (IQR) with whiskers extending to 1.5 times the IQR. The blue line represents the median, while the red diamond indicates the mean.

Figure 3: Performance of GoA under different retrieval methods with Llama 3.1 8B with 2K context window on Qasper, HotpotQA, and 2WikiM. $x$-axis represents different methods. $y$-axis represents average performance change from three different random seeds. The marker is the average change with the bar represent the standard error.

### 5.3 GoA's COMPRESSION BEST DESCRIBES THE QUESTION

We now evaluate whether GoA's compression strategy, based on greedy optimization of embedding similarity, successfully generates summaries that are maximally informative about the query. Figure 2 shows the cosine similarity between the final compressed summary and the query for each method, providing strong evidence for our hypothesis. Specifically, GoA consistently produces summaries that are most semantically aligned with the query. It achieves the highest mean and median similarity and, crucially, the lowest variance, confirming that our greedy optimization algorithm effectively maximizes the intended objective. In contrast, the baselines reveal their structural weaknesses. RAG's lower similarity demonstrates the limitations of simple filtering, which cannot synthesize disparate information into a holistically relevant summary. CoA's high variance highlights the pitfalls of its static left-to-right collaboration structure: its performance degrades significantly when a document's key information is not organized linearly. This analysis confirms that GoA's input-dependent collaboration is key to generating high-quality summaries.

### 5.4 SEMANTIC, CONTEXTUAL SEARCH IS CRUCIAL FOR GoA

Figure 3 confirms that semantic, contextual search that adaptively selects the next chunk *semantically* closest to the question *given* the previous context (cf. (7)) is a key component of GoA. *Semantic search*: Replacing semantic search with lexical BM25 leads to a larger performance degradation than replacing it with another semantic search method (ColBERT). This confirms that GoA's performance is tied to its ability to identify semantically relevant information, as intended by our algorithm design (cf. §4.2). *Contextual search*: Ablating contextual search (choosing the next chunk based only on its similarity to the query without the intermediate summary) leads to a severe degradation in performance. Without context, GoA no longer optimizes the principled objective (2), which can result in an incoherent and repetitive final summary.

### 5.5 SENSITIVITY ANALYSES

In Appendix C, we carefully analyze algorithmic design choices of GoA with a series of sensitivity analyses with Llama 3.1 8B with 2K context window on Qasper, HotpotQA, and 2WikiM. Findings can be summarized as: *GoA's performance is robust to the choice of document embedding models (Figure A4).* However, switching the embedding model by the average of GloVe word embedding (Pennington et al., 2014) causes a significant performance drop, which weakens a connection to the distributional similarity as opposed to the InfoNCE-trained sentence embedding models. *$k$-mediod clustering turns out to be suboptimal, being dominated by $k$-means and spectral clustering (Figure A4).* A plausible explanation is that our dataset is either less influenced by outliers than anticipated or exhibits a complex, non-globular structure more effectively captured by spectral clustering. *The number of subgraphs reveals a key trade-off (Figure A5),* achieving the

best score at $k = 2$ and $k = 4$ for Llama and Qwen, respectively. Too much parallelism (e.g., $k = 8$ as in without contextual compressors) creates chains too short for meaningful context, while too little (e.g., $k = 1$ as in CoA) results in a single long chain that "forgets" early information.

## 6 RELATED WORK

**Multi-Agent Systems.** The direction closest to ours is multi-agent system-based approaches that leverage multiple LLM agents collaboratively solve complex tasks with long context inputs. A common paradigm in these systems is to define agents with specialized roles and orchestrate their interactions through a pre-defined workflow. For instance, agents may be specialized for atomic actions like question decomposition and fact verification (Zhao et al., 2025), or a supervising agent may select and manage other task-specific agents (e.g., for summarization or QA) (Zhao et al., 2024a). Many other systems follow similar patterns of role specialization and fixed interaction protocols (Zhang et al., 2025a; Chen et al., 2024b; Zhang et al., 2024; Gur et al., 2024; Li et al., 2024; Sun et al., 2023). While often effective, these approaches rely on pre-defined collaboration strategies that may not be effective for every problem instance. In contrast, rather than prescribing fixed roles and interactions, GoA employs homogeneous worker agents guided by a fundamental objective for long context modeling (cf. (2)). This allows the optimal collaboration strategy for summarizing the input to emerge dynamically from the statistical structure of each query and context.

**Input Reduction Approaches.** GoA is also conceptually related to approaches aiming for reducing long context input. RAG is arguably the most widely used approach in this category due to its simplicity and strong empirical performances (see survey (Gao et al., 2023) for more details). However, RAG's filtering mechanism is inherently incapable of exploiting interdependencies across the entire context, limiting its effectiveness on tasks that require complex reasoning. To address this, subsequent works have employed LLMs as more sophisticated compressors. These methods may progressively summarize chunks into a tree structure (Chen et al., 2023a), maintain a compressed "gist memory" for retrieval (Lee et al., 2024), or use LLMs to filter redundant sentences (Jin et al., 2025). While an improvement over simple filtering, these approaches typically perform compression locally on isolated chunks, lacking a global objective to guide the synthesis of information. GoA addresses this shortcoming directly: its principled objective guides agents to construct a globally coherent summary that is maximally informative for the given query.

## 7 CONCLUSION & FUTURE WORK

We introduce a principled information-theoretic perspective on long context modeling, recasting it as a compression problem. This framing yields a mutual information objective that guides the design of multi-agent systems, moving beyond heuristic approaches. We propose GoA, a framework that operationalizes this objective by dynamically constructing an input-dependent collaboration graph. Unlike prior methods that rely on a static hand-crafted collaboration structure, GoA's emergent structure is tailored to each input and query. As a result, GoA significantly extends the effective context window of LLMs, demonstrably outperforming strong retrieval-based and static multi-agent baselines.

We remark several promising future direction that can further generalize a flexible descriptive approach to multi-agent collaboration for long context modeling: *Generalizing the Collaboration Graph*: The linear forest structure in GoA could be generalized to more complex topologies, such as directed acyclic graphs, to enable richer information flow. Furthermore, the disjoint partitioning of chunks could be relaxed to allow key information to act as a shared context between subgraphs, albeit with the expense of repeating inference on the shared context. *Learning to Collaborate*: While we present GoA as a training-free method, its information-theoretic objective provides a natural loss function for learning collaboration. This could involve training a policy to construct the optimal graph or fine-tuning agents for query-aware compression to directly maximize the objective in Eq. (2).

## REPRODUCIBILITY STATEMENT

We attach the code used in this paper, including the implementation of the Graph of Agents framework and baselines, and will open-source these libraries. Implementation details can be found in Appendix E.

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

## A    PROOFS OF CLAIMS

### A.1    PROOF OF PROPOSITION 1

**Proposition 1.** *For the Bayes optimal predictor $\hat{P}$, the following equivalence relationship holds:*

$$\mathbb{E}_{(X,Q,Y)\sim P}\left[\log \hat{P}(Y|g(X),Q)\right] \quad \Longleftrightarrow \quad I(Y; g(X), Q), \tag{2}$$

*where $I(X;Y) \triangleq \mathbb{E}\left[\log \frac{P(X,Y)}{P(X)P(Y)}\right]$ is the mutual information.*

*Proof.* The Bayes optimal predictor $\hat{P}$ satisfies $\hat{P}(Y|g(X),Q) \overset{a.e.}{=} P(Y|g(X),Q)$ (Lemma 3.1 in (Jeon et al., 2024)). Then, we get:

$$\begin{aligned} \mathbb{E}_{(X,Q,Y)\sim P}\left[\log \hat{P}(Y|g(X),Q)\right] &= \mathbb{E}_{(X,Q,Y)\sim P}\left[\log P(Y|g(X),Q)\right] \\ &= -H(Y|g(X),Q) + H(Y) - H(Y) \\ &= I(Y; g(X), Q) - H(Y). \end{aligned}$$

The equivalence result follows from the constant nature of $H(Y)$ with respect to $g$. $\qquad\square$

### A.2    PROOF OF EQUIVALENCE

We first formalize the observation in (Liu et al., 2025) about a strong correlation between the log odds ratio of the answer $\log \frac{P(y|q,x)}{1-P(y|q,x)}$ and the pointwise mutual information $PMI(q,x) \triangleq \log \frac{P(q,x)}{P(q)P(x)}$.

**Observation 1** (Generalized version of Hypothesis 2.1 in Liu et al. (2025)). *For any choices of $(x,q,g)$, there exist constants $\alpha > 0$ and $\beta$ such that*

$$\log \frac{P(y|q,g(x))}{1 - P(y|q,g(x))} = \alpha PMI(q, g(x)) + \beta + \epsilon(q,x), \tag{8}$$

*where $y$ is an answer to the query $q$ and the input $x$ and $\epsilon(q,x)$ is a zero-mean residual uncorrelated with the choice of $g$.*

We establish the equivalence between maximizing $I(Y; g(X), Q)$ and the log odds ratio of the answer with respect to $g$ as

$$I(Y; g(X), Q) \triangleq \mathbb{E}\left[\log \frac{P(Y|Q, g(X))}{P(Y)}\right]$$

$$\Longleftrightarrow \mathbb{E}\left[\log P(Y|Q, g(X))\right] \Longleftrightarrow \mathbb{E}\left[\log \frac{P(Y|Q, g(X))}{1 - P(Y|Q, g(X))}\right] \tag{9}$$

where the first equivalence is obtained by removing the term constant with respect to $g$ and the second equivalence is due to the monotonicity.

Under Observation 1 above, for each observed triple $(y,q,x,g)$, we get

$$\log \frac{P(y|q,g(x))}{1 - P(y|q,g(x))} = \alpha PMI(q, g(x)) + \beta + \epsilon(q,x). \tag{10}$$

Taking expectation both sides gives

$$\mathbb{E}\left[\log \frac{P(Y|Q, g(X))}{1 - P(Y|Q, g(X))}\right] = \alpha \mathbb{E}[PMI(q, g(x))] + \beta, \tag{11}$$

where maximizing the left-hand side over $g$ is equivalent to maximizing $\mathbb{E}[PMI(q, g(x))]$.

Finally, recall the identity $\mathbb{E}[PMI(Q, g(X))] = I(Q; g(X))$, so we connect $\max_g I(Y; g(X), Q)$ with $\max_g I(Q; g(X))$ by combining (9) and (11).

## B  ADDITIONAL THEORETICAL RESULT

**Proposition 2.** *Suppose LLM $\hat{P}$ has a uniform bound for the expected error $\mathbb{E}_{(X,Q,Y)\sim P}[D_{\mathrm{KL}}(P(Y|g(X),Q))||\hat{P}(Y|g(X),Q))] \leq \epsilon$ for any compressor $g$. Consider two compression functions $g_1$ and $g_2$ such that $I(Y; g_1(X), Q) > I(Y; g_2(X), Q) + \delta$ for $\delta > 0$. Then if $\delta > \epsilon$, it holds that:*

$$\mathbb{E}_{(X,Q,Y)\sim P}\left[\log \hat{P}(Y|g_1(X),Q)\right] > \mathbb{E}_{(X,Q,Y)\sim P}\left[\log \hat{P}(Y|g_2(X),Q)\right]. \tag{12}$$

*Proof.* For any compression function $g$ and predictor $\hat{P}$, we can decompose the expected log-likelihood as

$$\mathbb{E}\left[\log \hat{P}(Y|g(X),Q)\right] = \mathbb{E}\left[\log P(Y|g(X),Q)\right] - \mathbb{E}[D_{\mathrm{KL}}(P(Y|g(X),Q))||\hat{P}(Y|g(X),Q))] \tag{13}$$

$$= I(Y; g(X), Q) - H(Y) - \mathbb{E}[D_{\mathrm{KL}}(P(Y|g(X),Q))||\hat{P}(Y|g(X),Q))] \tag{14}$$

where (14) is based on Proposition 1. Here, all expectations are with respect to $(X, Q, Y) \sim P$ and omitted for brevity.

Then, for $g_1$ and $g_2$ such that $I(Y; g_1(X), Q) > I(Y; g_2(X), Q) + \delta$ for $\delta > 0$, we get the desired result as:

$$\mathbb{E}\left[\log \hat{P}(Y|g_1(X),Q)\right] - \mathbb{E}\left[\log \hat{P}(Y|g_2(X),Q)\right]$$
$$= I(Y; g_1(X), Q) - I(Y; g_2(X), Q)$$
$$\quad - \mathbb{E}\left[D_{\mathrm{KL}}(P(Y|g_1(X),Q)|\hat{P}(Y|g_1(X),Q)) - D_{\mathrm{KL}}(P(Y|g_2(X),Q)|\hat{P}(Y|g_2(X),Q))\right]$$
$$> \delta - \epsilon > 0.$$

$\square$

## C  ABLATION STUDY & SENSITIVITY ANALYSIS

*GoA's performance is robust to external models.* As shown in Figure A4, GoA is robust to the choice of modern embedding models (BGE-M3 vs. Qwen3 Embedding (Zhang et al., 2025b)). However, switching to the average of GloVe word embedding (Pennington et al., 2014) causes a significant performance drop. This could be contributed to its weak connection to the distributional similarity as opposed to the InfoNCE-trained sentence embedding models.

*k-mediod clustering turns out to be suboptimal.* Surprisingly, we found that $k$-means and spectral clustering outperformed our initial choice of $k$-medoids. In the initial phase of our work, we opted to choose the $k$-medoid algorithm since it is less sensitive to outliers and operates on actual data points as cluster representatives, enabling representative-based search strategies. In this regard, a plausible explanation is that our dataset is either less influenced by outliers than anticipated or exhibits a complex, non-globular structure more effectively captured by spectral clustering.

*The number of subgraphs reveals a key trade-off.* Figure A5 shows a clear hill-shaped curve for performance as $k$ changes, achieving the best score at $k = 2$ and $k = 4$ for Llama and Qwen, respectively. This illustrates

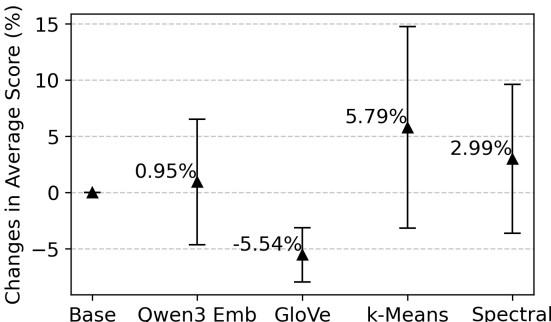

Figure A4: Performance of GoA under different embedding and clustering methods with Llama 3.1 8B with 2K context window on Qasper, HotpotQA, and 2WikiM. $x$-axis represents different methods. $y$-axis represents average performance change from three different random seeds. The marker is the average change with the bar represent the standard error.

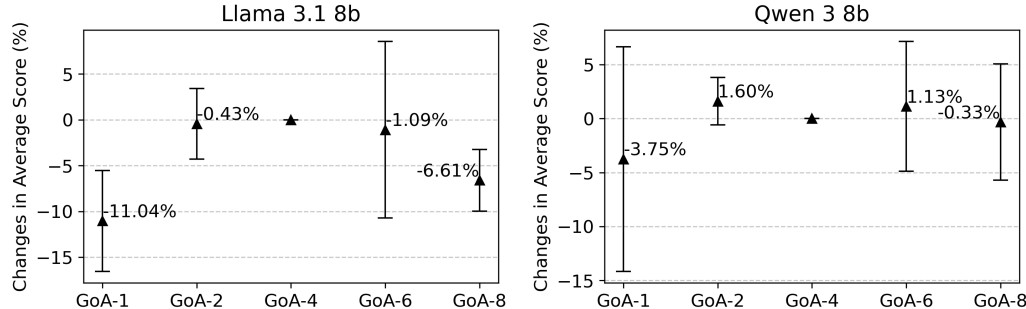

Figure A5: Performance of GoA under different number of subgraphs with Llama 3.1 8B with 2K context window on Qasper, HotpotQA, and 2WikiM. In $x$-axis, GoA-$k$ represents GoA with $k$ number of subgraphs. $y$-axis represents average performance change from three different random seeds. The marker is the average change with the bar represent the standard error.

the fundamental trade-off between context length and parallelism. For large $k$, each chain is too short to build meaningful context, degrading performance at the expense of supporting massive parallelization. Conversely, for $k = 1$, GoA becomes a single, long chain similar to CoA. The significant performance drop here highlights the "forgetting phenomenon" in long-chain models: information from early chunks is progressively diluted in the iterative summarization process. GoA's use of shorter, parallel chains ($k > 1$) effectively mitigates this issue.

## D    DISCUSSION

The per-sample collaboration in GoA generalizes the static left-to-right communication in CoA, suggesting GoA's superiority when semantic and distributional similarities are equivalent (cf. §4.2) and the greedy algorithm is effective (cf. §4.3). However, it remains unclear how GoA performs against other model-agnostic approaches like RAG and multi-agent systems. To investigate this, we analyze the implicit compression function within these existing methods. To this end, we formalize GoA and CoA

as a contextual compressor $g^{\text{ctx}}(\texttt{chunk};\texttt{context})$, where prior information is used to compress subsequent chunks. For example, the CoA compression scheme (a special case of GoA) can be expressed as: $\tilde{x}^{\text{ctx}} = [g^{\text{ctx}}(c_1), g^{\text{ctx}}(c_2; c_1), \cdots, g^{\text{ctx}}(c_l; c_1, \cdots, c_{l-1})]$.

**Multi-Agent Systems without Contextual Compressors.** Despite differences in prompts and overall communication protocols, multi-agent systems involve an LLM-based compressor that operates for separate chunks (e.g., the collaborative reasoning steps in Zhao et al. (2024a)), with the notable exception of CoA and multi-agent debates (Wynn et al., 2025). This chunk-wise compression through an LLM, denoted as $g^{\text{Agent}}$, can be formalized as

$$\tilde{x}^{\text{Agent}} \triangleq g^{\text{Agent}}(x) = [g^{\text{Agent}}(c_1), g^{\text{Agent}}(c_2), \cdots, g^{\text{Agent}}(c_l)]. \tag{15}$$

The lack of contextual information in $g^{\text{Agent}}$ makes it fundamentally inferior to $g^{\text{ctx}}$ in maximizing the compression objective $I(Y; g(X), Q) = I(Y; g(C_1), Q) \sum_{i=2}^{l} I(Y; g(C_i), Q | g(C_1), \cdots, g(C_{i-1}))$ because it discards context before compression is applied. Specifically, it creates an irreversible information bottleneck from the Markov chain $Y \rightarrow (C_1, C_2, \cdots, C_l) \rightarrow C_i \rightarrow g^{\text{agent}}(C_i)$. Consequently, this approach is limited to optimizing $I(Y; g^{\text{agent}}(C_i), Q | C_1, \ldots, C_{i-1})$ for each chunk, falling short of the contextual compressor's objective, which includes the prior context $(C_1, \ldots, C_{i-1})$ within the compression function itself. This loss of context prevents the compressor from performing critical functions like disambiguating terms or eliminating redundancy.

**RAG.** RAG involves an implicit compression algorithm $g^{\text{RAG}}$ that retains all information if a chunk is among the top-$\kappa$ relevant chunks, and otherwise removes it entirely: $\tilde{x}^{\text{RAG}} \triangleq g^{\text{RAG}}(x) = [g^{\text{RAG}}(c_1), \cdots, g^{\text{RAG}}(c_l)]$ where

$$g^{\text{RAG}}(c_i) = \begin{cases} c_i & \text{if } c_i \in \text{Top-}\kappa \text{ relevant chunks among } \{c_j\}_{j=1}^{l} \\ \emptyset & \text{otherwise} \end{cases}. \tag{16}$$

The compression algorithm $g^{\text{RAG}}$ has severe structural limitations compared to $g^{\text{ctx}}$ and $g^{\text{Agent}}$. Specifically, the filtering-style compression in RAG makes it inherently hard to understand the entire context or capture interdependency between multiple chunks, leading to its deficient performance on complex tasks such as summarization and multi-hop reasoning. Besides, from the perspective of empirical laws on diminishing returns in mutual information, which exhibit logarithmic information gains with more tokens, RAG does not exploit initial sharp increases in mutual information that could have been obtained by just retaining few tokens from each chunk.

# E   ADDITIONAL DETAILS

## E.1   IMPLEMENTATION DETAILS

For all experiments, we adopt a generic setting to minimize biases from dataset-specific configurations. We use a decoding temperature of 0.1 and a top-p of 0.9. For manager LLMs and vanilla LLMs that produce the final answer, the maximum number of output tokens is set to 128. For worker LLMs, the maximum number of output tokens is set to 256 for models with a 2K context window and 1024 for models with an 8K context window. When an input's length exceeds a model's context limit, we truncate the middle of the input, a strategy shown to have the least performance degradation (Bai et al., 2024). The final answer is extracted from the generated text enclosed within `<answer>ExtractedAnswer</answer>` tags. Specific prompts used are detailed in §E.2 and §E.3.

### E.2 PROMPTS FOR VANILLA LLMS

The prompts for the baseline LLM evaluations are adapted from LongBench (Bai et al., 2024). We remove all task-specific instructions, preserving only the guidelines for output formatting. This foundational prompt structure is also used for our Graph of Agents (GoA) method and the Chain-of-Agents (CoA) baseline.

---

**Vanilla LLM Prompt (Multi-Document QA).**

```
Answer the question based on the given passages. Only give me the answer
↪  and do not output any other words. Put your answer inside the answer
↪  tag, like <answer>your answer</answer>.

The following are given passages.
{context}

Answer the question based on the given passages. Only give me the answer
↪  and do not output any other words. Put your answer inside the answer
↪  tag, like <answer>your answer</answer>.

Question: {input}
Answer:
```

---

**Vanilla LLM Prompt (Single-Document QA).**

```
Answer the question based on the given passages as concisely as you can,
↪  using a single phrase or sentence if possible. If the question
↪  cannot be answered based on the information in the given passages,
↪  write "unanswerable". If the question is a yes/no question, answer
↪  "yes", "no", or "unanswerable". Do not provide any explanation. Put
↪  your answer inside the answer tag, like <answer>your
↪  answer</answer>.

The following are given passages.
{context}

Answer the question based on the given passages as concisely as you can,
↪  using a single phrase or sentence if possible. If the question
↪  cannot be answered based on the information in the given passages,
↪  write "unanswerable". If the question is a yes/no question, answer
↪  "yes", "no", or "unanswerable". Do not provide any explanation. Put
↪  your answer inside the answer tag, like <answer>your
↪  answer</answer>.

Question: {input}

Answer:
```

### E.3 PROMPTS FOR COA AND GOA

The prompts for CoA and GoA are largely identical. The worker prompts for both methods are intentionally generic, instructing the agent to summarize the input text based on the query to ensure broad applicability. The manager prompt, which generates the final answer, is the same for both methods as well. It uses the vanilla LLM prompt structure but adds the instruction: `However, the source text is too long and has been summarized. You need to generate a final summary based on the provided summary` (Zhang et al., 2024). The only distinction arises in how GoA structures the input for the manager agent. Because GoA generates summaries from multiple subgraphs, these are concatenated and presented to the manager, with each summary section clearly marked by the header `[Summary of Worker i out of k]`.

---

**Worker LLM Prompt.**

```
You need to read [SOURCE TEXT] and [PREVIOUS SUMMARY] and generate a
↪   summary to include them both. Later, this summary will be used for
↪   other agents to answer [QUERY], if any. So please write the summary
↪   that can include the evidence for answering [QUERY].

[SOURCE TEXT]: {input_chunk}

[PREVIOUS SUMMARY]: {prev_cu}

[QUERY]: {query}

Summary:
```

---

**Manager LLM Prompt (Multi-Document QA).**

```
Answer the question based on the provided information. Only give me the
↪   answer and do not output any other words. Put your answer inside the
↪   answer tag, like <answer>your answer</answer>.

The following are given passages. However, the source text is too long
↪   and has been summarized. You need to generate a final summary based
↪   on the provided summary:

{summary}

Answer the question based on the given passages. Only give me the answer
↪   and do not output any other words. Put your answer inside the answer
↪   tag, like <answer>your answer</answer>.

Question: {query}

Answer:
```

```
Manager LLM Prompt (Single-Document QA)

Answer the question based on the given passages as concisely as you can,
↪   using a single phrase or sentence if possible. If the question
↪   cannot be answered based on the information in the given passages,
↪   write "unanswerable". If the question is a yes/no question, answer
↪   "yes", "no", or "unanswerable". Do not provide any explanation. Put
↪   your answer inside the answer tag, like <answer>your
↪   answer</answer>.

The following are given passages. However, the source text is too long
↪   and has been summarized. You need to generate a final summary based
↪   on the provided summary:

{summary}

Answer the question based on the given passages as concisely as you can,
↪   using a single phrase or sentence if possible. If the question
↪   cannot be answered based on the information in the given passages,
↪   write "unanswerable". If the question is a yes/no question, answer
↪   "yes", "no", or "unanswerable". Do not provide any explanation. Put
↪   your answer inside the answer tag, like <answer>your
↪   answer</answer>.

Question: {query}

Answer:
```

## E.4 DATASET DETAILS

Below we give a brief background on each of the datasets in our evaluations, and in Table A2 we list summary statistics showing the context length in terms of the number of tokens in samples from the datasets. In Figure A6 we plot the distribution of token lengths for samples within each of the six Longbench v1 datasets.

**Single Doc QA Datasets** In NarrativeQA (Kočiský et al., 2018) the task is to answer questions about stories or scripts. The questions require understanding elements such as characters, plots, and themes. The Qasper (Dasigi et al., 2021) dataset about NLP research papers, where NLP practitioners proposed the questions and answers. MultiFieldQA (Bai et al., 2024) consists of data from long articles from 10 sources, including Latex papers, judicial documents, government work reports, and PDF documents indexed by Google. For each article, several graduate students annotated questions and answers.

**Multi Doc QA Datasets** The Hotpot (Perez et al., 2020) dataset requires reasoning across multiple passages to find an answer, where questions are sourced from Wikipedia. 2WikiM (Ho et al., 2020) requires the system to answer questions based on multiple given documents, which requires finding a reasoning path across the multi-hop questions. MuSiQue (Trivedi et al., 2022) is a multi-hop question and answering dataset that is more difficult than HotpotQA (of which it is based on). The MuSiQue dataset contains more hops per sample, questions without answers, and additional distracting content.

Table A2: Summary statistics on the token lengths of context for each evaluation dataset. Datasets are tokenized with Llama 3.1 8B's tokenizer to find sequence lengths.

|  | Single Doc QA | | | Multi Doc QA | | |
|  | NarrativeQA | Qasper | MultifieldQA | Hotpot | 2WikiM | Musique |
| --- | --- | --- | --- | --- | --- | --- |
| Median | 31284 | 4536 | 6947 | 14208 | 6280 | 16106 |
| Mean | 29776 | 4921 | 6888 | 12779 | 7096 | 15542 |
| St. Dev | 17271 | 2603 | 3408 | 3772 | 3469 | 1567 |
| Minimum | 7963 | 1846 | 1286 | 1726 | 912 | 6482 |
| Maximum | 65270 | 21110 | 14947 | 16322 | 16319 | 16335 |

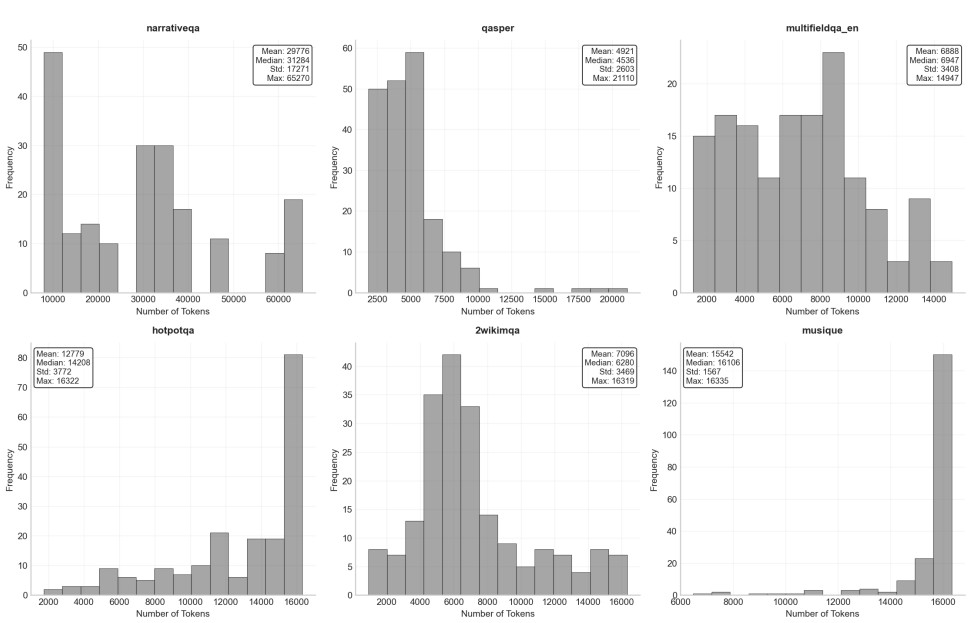

Figure A6: Distribution of context lengths for each dataset within Longbench v1, as measured by number of tokens. Texts are tokenized using the Llama 3.1 tokenizer.