# OpenReview forum: "Graph of Agents: Principled Long Context Modeling by Emergent Multi-Agent Collaboration"
_ICLR.cc/2026/Conference — Submitted to ICLR 2026_

### Official Review · Reviewer_ZUtv · 2025-10-26

**Soundness:** 2
**Presentation:** 3
**Contribution:** 3
**Rating:** 4
**Confidence:** 4

**Summary:**

This paper introduces Graph-of-Agents (GoA), a method that begins by semantically clustering text chunks. For each cluster, it then constructs a path graph by assessing the mutual information between the chunks and the specific input question. An agent component sequentially summarizes the documents along this path graph. Finally, the aggregated summary is passed to a Large Language Model to generate the answer.

**Strengths:**

The core strength of this work lies in its input-dependent graph construction. From a compression perspective, this approach dynamically measures the relevance between the question and the incrementally built summary, aiming to achieve efficient information compression and long-context question answering.

**Weaknesses:**

1. Despite the "Graph" nomenclature, the resulting structure is a "path graph"—a linear list. This architecture is structurally similar to a "Chain of Agents," which significantly diminishes the novelty and the claimed advantages over simpler, chain-based methods.

2. The path graph construction process appears computationally intensive. Even with a greedy algorithm, adding each new node incurs a computational complexity of o(N), where N is the total number of chunks. This overhead could be very high for long contexts.

3. The algorithm's objective function is based on the similarity between the LLM's summary and the question. However, the underlying assumptions seem overly idealized, casting doubt on the method's practical feasibility. For instance, LLM may inevitably omit critical, question-relevant details during its summarization pass. Furthermore, GoA also does not account for cases requiring the synthesis of information from multiple, non-sequential chunks.

4. The entire graph is question-dependent and offers no reusability. A new graph must be constructed de novo for every single query, which severely limits the method's practicality and scalability in real-world applications.

**Questions:**

N/A

---

> ### Author Response · Authors · 2025-12-03
>
> We thank the reviewer for the thoughtful feedback. We address the specific concerns below:
>
>
> # W1
>
> We clarify that GoA constructs a linear forest (a collection of disjoint paths), not a single linear list. Further, unlike Chain-of-Agents (CoA) or fixed-order methods, GoA dynamically determines the optimal processing order to cluster semantically related chunks. This dynamic structuring is key to our performance gains in multi-document understanding.
>
> # W2
>
> While graph construction adds a step, GoA can reduce total latency through parallelization. Unlike CoA, which requires strictly sequential processing of all chunks, GoA's linear forest structure allows multiple paths to be processed concurrently. This parallel execution offsets the construction overhead, making it efficient for long contexts.
>
> # W3
>
> We acknowledge the hypothetical concern regarding potential information loss during summarization. However, our empirical results demonstrate that the mutual information-based objective effectively retains critical details. The experimental performance confirms that the algorithmic design is robust against the information loss concerns raised.
>
>
> # W4
>
> While the graph is constructed per query, we utilize a small language model (e.g., BERT-based model) for this process. Consequently, the computational cost of graph construction is negligible compared to the LLM chunk processing, ensuring the method remains scalable in real-world applications.

---

### Official Review · Reviewer_D7yF · 2025-10-31

**Soundness:** 3
**Presentation:** 3
**Contribution:** 3
**Rating:** 4
**Confidence:** 4

**Summary:**

This paper reframes model-agnostic long-context modeling as a compression problem and derives a principled objective via mutual information, then instantiates it with GoA—a dynamic, input-dependent linear forest of collaborating worker LLMs plus a manager. The graph is built by clustering semantically related chunks and greedily ordering within each cluster to maximize semantic alignment between the evolving summary and the query, motivated by InfoNCE/PMI connections. Empirically, on six LongBench QA tasks with Llama-3.1-8B and Qwen3-8B, GoA improves averaged F1 over strong RAG and Chain-of-Agents baselines and, notably, with a 2K window surpasses Llama-3.1-8B with a 128K window. Figures and Table 1 support these gains and ablations show the importance of semantic, contextual next-chunk selection.

**Strengths:**

* This paper is clear writing and easy to follow.

* Across two backbones and six datasets, GoA consistently matches or beats RAG/CoA at the same context length; with 2K context it even edges the 128K vanilla model

**Weaknesses:**

* Experiments focus on LongBench QA; claims about “long-context modeling” could be tested on repo-level code understanding, or agentic tasks to assess generality of the compressor.

* While linear forests allow parallelism, the paper does not quantify total token usage, wall-clock latency, or budget vs. RAG/CoA under equal compute constraints. (Algorithm suggests multiple LLM calls.)

* Appendix shows k-medoids underperforms k-means/spectral; the main results fix 𝑘=4 and a particular clustering, leaving open how robust performance is under automatic selection of 𝑘/method.

**Questions:**

* Did you try simple look-ahead (beam) or submodularity analyses to bound the greedy gap? Even small beams could test whether myopia matters in multi-hop cases.

* Under a fixed token/latency budget, how do GoA, RAG (with reranking), and CoA compare in end-to-end wall-clock and dollar cost, given multiple worker calls + manager? A cost-efficiency plot would be valuable.

---

> ### Author Response · Authors · 2025-12-03
>
> We thank the reviewer for the valuable comments. We address the specific points below:
>
>
> # W1
>
> We agree that applying GoA to agentic tasks and repository-level code understanding is a promising direction. However, this work focuses on establishing the theoretical and empirical foundations of GoA for long-context QA. We leave the extension to complex agentic workflows and code environments for future work.
>
> # W2
>
> We emphasize that GoA can offer a significant latency advantage through parallelization. While Chain-of-Agents (CoA) is restricted to a strictly sequential linear chain, GoA constructs a linear forest. This allows independent paths to be processed in parallel, significantly reducing wall-clock time compared to sequential methods.
>
>
> # W3
>
> We selected the k-medoids algorithm specifically to enable the representative-based search strategies. Even with this choice, GoA consistently outperforms strong baselines (RAG, CoA). This demonstrates that the framework's effectiveness is robust and not contingent on using the absolute optimal clustering method (e.g., k-means/spectral).
>
>
> # Q1
>
> To ensure a fair and controlled comparison across all baselines, we standardized the decoding and search strategies. We focused on the greedy approach to demonstrate the effectiveness of the core GoA architecture without introducing the confounding variables of more complex search heuristics like beam search.
>
>
>
> # Q2
>
> Regarding the trade-off between cost and speed, we highlight that GoA prioritizes latency reduction. By structuring the context as a linear forest rather than a single chain, GoA leverages parallel processing to achieve faster inference times than CoA, making it a more time-efficient solution for long-context modeling.

---

### Official Review · Reviewer_EDBe · 2025-11-01

**Soundness:** 2
**Presentation:** 2
**Contribution:** 1
**Rating:** 2
**Confidence:** 3

**Summary:**

In this paper, the authors propose Graph-of-Agents (GoA), a framework for long context question answer tasks. In GoA, the input text is first split into chunks, then clustered according to their embedding. Next, each worker agent working on each cluster is given these chunks and is asked to summarize them in order of similarity of the chunk and the current summary and query. Finally, a manager agent reads the results and generates the answer. Results on 6 datasets with 2 models show that the proposed method surpasses RAG and CoA baselines on average.

**Strengths:**

1. Long context task is an important and urgent issue in LLMs.
2. The results show that the average performance is better than baseline.

**Weaknesses:**

1. The biggest concern is novelty and contribution. Compared with the baseline, the new component is basically clustering the chunks of document into different groups and assigning them to multiple Worker agents. However, such a method is widely used in various papers (document clustering works). Besides, cluster-then-process only changes the reading order of the baseline method, which it difficult to recognize it as a significant contribution of the method.

2. The improvement is unjustified. Since many results are lower than the baseline, it is important to carefully conduct an analysis of why it is lower. On the other hand, some case studies on why the proposed method is better than the baseline are also needed. Without these analyses, it is not very convincing to believe the proposed method works well.

3. The experiments are not solid. Some baselines can be added to enhance the results. Such as LongAgents, which is also a tree structure. Also, it is not very clear about the effectiveness of each component in the framework, such as an ablation study for random order GoA or random clustered GoA. Furthermore, the motivation for using clustering of documents can be tested, for instance, whether clustering can break the natural order of the document, and it is not certain if there really exist some topics for each cluster, or just a mixture of things in each chunk.

**Questions:**

See above.

---

> ### Author Response · Authors · 2025-12-03
>
> We thank the reviewer for the helpful feedback. We address the concerns regarding novelty and experiments below:
>
> # W1
>
> We respectfully disagree that the contribution is trivial. Mathematically, determining the optimal information processing order to maximize compression is an intractable optimization problem. Our core novelty lies in formulating this as a linear forest construction problem and providing a principled, efficient approximation algorithm based on mutual information, rather than relying on arbitrary heuristics or fixed sequential reading.
>
>
> # W2
>
> Our analysis indicates that GoA is particularly dominant in Multi-Document QA, where information is scattered without a natural sequential order. While sequential methods may benefit from the narrative flow in single-document tasks, GoA’s ability to dynamically reorder and cluster disjoint information is critical for multi-document understanding, validating the graph-based approach.
>
>
> # W3
>
> Regarding LongAgents, we excluded it because it requires additional model training and specialized role-based prompts. In contrast, GoA and the selected baselines are training-free, inference-time methods, making them a more direct comparison. Regarding ablations, we refer the reviewer to Sections 5.4 and 5.5, where we explicitly analyze the impact of clustering and ordering (including random baselines), demonstrating that our semantic strategies significantly outperform random permutations.

---

### Official Review · Reviewer_8Ngo · 2025-11-01

**Soundness:** 3
**Presentation:** 3
**Contribution:** 3
**Rating:** 4
**Confidence:** 4

**Summary:**

The paper proposes Graph of Agents (GoA), a multi-agent framework for long-context reasoning in LLMs. Instead of a fixed Chain of Agents (CoA)—a serial pipeline where each agent summarizes one chunk—GoA dynamically constructs a graph (specifically, a linear forest) of agents based on semantic similarity among chunks. Each subgraph processes a semantically coherent subset of text, and all run in parallel; their outputs are later combined by a manager agent. Theoretical grounding is provided: for a Bayes-optimal predictor, maximizing expected log-likelihood equals maximizing mutual information between compressed input and answer, motivating semantic information preservation as the key compression criterion. Empirically, GoA outperforms baselines (vanilla, RAG, CoA) on LongBench QA tasks using small-context LLMs, suggesting improved effective context length. However, dynamic graph construction introduces non-trivial overhead not measured in latency terms, leaving open the practical speed-accuracy trade-off.

**Strengths:**

+ principled design: it derives the agent architecture from an information-theoretic formulation rather than heuristics. The proposed GoA generalizes prior CoA structures, introducing dynamic, input-dependent collaboration among agents. This enables better modeling of non-linear information dependencies in long documents.
+ Empirical results demonstrate consistent accuracy improvements across six QA datasets, even outperforming models with much larger context windows. The authors also provide meaningful ablations (embedding models, clustering strategies, contextual vs. lexical search) that validate design choices. The method’s modularity makes it model-agnostic—applicable to any base LLM without fine-tuning—and it is theoretically elegant in unifying mutual-information maximization with graph-structured collaboration.

**Weaknesses:**

- GoA’s performance depends heavily on the semantic embedding and clustering quality; if embeddings fail to capture query relevance, summarization paths degrade.

- The dynamic graph construction—computing embeddings, clustering, building subgraphs, and launching multiple agent calls—introduces computational overhead. Now in section 4.1, the authors claim that Parallelization "significantly
reducing latency". I find this hard to believe for the reason mentioned. For this claim, I would encourage the authors to to report latency or time-to-first-token measurements.

**Questions:**

Proposition 1 Motivation: The Bayes-optimal predictor result equating log-likelihood and mutual information seems mainly illustrative. Is there a deeper analogy to be interpreted?

Clarification: Section 5.1 datasets mentions "The datasets range from 3.6K to 18K tokens,..", but table-1 references 128-K?

---

> ### Author Response · Authors · 2025-12-03
>
> We thank the reviewer for the insightful comments. We address the specific concerns below:
>
>
>
> # W1
>
> We acknowledge the dependency of GoA on the qualities of semantic embedding and clustering, but our sensitivity analysis in Section 5.3 demonstrates empirical robustness. As shown in Figure 3 and Figure A4, replacing the semantic embedding models or clustering algorithms with similarly performant alternatives results in only mild performance fluctuations, indicating that GoA is not brittle to specific component choices.
>
> # W2
>
> We clarify that the primary latency bottleneck in these systems is the sequential processing of LLM chunks, not the lightweight graph construction by a small language model (e.g., BERT-based model). While Chain-of-Agents (CoA) forces a strictly serial execution, GoA's linear forest structure allows entire chains of chunks to be processed concurrently. This parallelization significantly outweighs the graph construction overhead, reducing total end-to-end latency.
>
> # Q1
>
> We clarify that Proposition 1 is not merely illustrative but a formal mathematical statement. It establishes that for a Bayes-optimal predictor, maximizing the predictive log-likelihood is mathematically equivalent to maximizing the mutual information between the compressed input and the target. This provides the rigorous theoretical foundation for our objective function.
>
> # Q2
>
> The "3.6K to 18K tokens" mentioned in Section 5.1 refers to the average document length of the datasets used. The "128K" in Table 1 refers to the maximum context window capacity of the baseline model (Llama-3.1-8B-128K) used for comparison, not the dataset size itself.

---

### Meta-Review · Area_Chair_7H2s · 2026-01-07

**Summary:**

The paper proposes Graph of Agents (GoA), a framework for long-context QA that clusters text chunks into a "linear forest" of agents to maximize information retrieval. Reviewers generally appreciate the principled information-theoretic formulation and the strong empirical results compared to RAG and Chain-of-Agents (CoA). However, there are significant reservations regarding the computational overhead of dynamic graph construction, the novelty of the clustering approach, and the absence of concrete latency/cost analysis. The consensus is that while the method performs well, the trade-off between accuracy and inference cost is not fully transparent.

**Reviewer Concerns:**

Addressed by Rebuttal:
- Structural Novelty: The authors successfully clarified to Reviewer ZUtv and EDBe that GoA constructs a "linear forest" (parallel paths) rather than a single sequential chain, distinguishing it from CoA.
- Robustness: Concerns regarding the brittleness of embedding choices (Reviewer 8Ngo) were addressed by pointing to sensitivity analyses already present in the paper.
- Baselines: The exclusion of LongAgents was reasonably justified (training-free vs. fine-tuned) in response to Reviewer EDBe.

Outstanding:
- Latency and Cost: Multiple reviewers (8Ngo, D7yF) requested hard data on wall-clock latency and token costs. While the authors argued theoretically that parallelization offsets graph construction overhead, they did not provide the specific measurements or cost-efficiency plots requested to prove this practical viability.
- Core Novelty: Reviewer EDBe's concern that the method is fundamentally just "clustering followed by summarization" remains largely subjective and arguably unaddressed, as this is the core mechanism of the paper.

**Reviewer Scores:**

Reviewer 8Ngo (4 → 4)

Reviewer EDBe (2 → 4)

Reviewer D7yF (4 → 4)

Reviewer ZUtv (4 → 4)

---

### Decision · Program_Chairs · 2026-01-26

Reject